# Population Pharmacokinetics of Prophylactic Cefazolin in Cardiac Surgery with Standard and Minimally Invasive Extracorporeal Circulation

**DOI:** 10.3390/antibiotics11111582

**Published:** 2022-11-09

**Authors:** Petr Šantavý, Martin Šíma, Ondřej Zuščich, Vendula Kubíčková, Danica Michaličková, Ondřej Slanař, Karel Urbánek

**Affiliations:** 1Department of Cardiac Surgery, Faculty of Medicine and Dentistry, Palacký University and University Hospital, 779 00 Olomouc, Czech Republic; 2Department of Pharmacology, First Faculty of Medicine, Charles University and General University Hospital, 128 00 Prague, Czech Republic; 3Department of Pharmacology, Faculty of Medicine and Dentistry, Palacký University and University Hospital, Hněvotínská 3, 775 15 Olomouc, Czech Republic

**Keywords:** cefazoline, nonlinear mixed-effects modeling, antibiotic prophylaxis, glomerular filtration rate, cardiopulmonary bypass

## Abstract

The objectives of this study were to develop a population pharmacokinetic model of prophylactically administered cefazolin in patients undergoing cardiac surgery with and without the use of the cardiopulmonary bypass of both existing types—standard (ECC) and minimallyu invasive extracorporeal circulation (MiECC)—and to propose cefazoline dosing optimization based on this model. A total of 65 adult patients undergoing cardiac surgery were recruited to this clinical trial. A prophylactic cefazolin dose of 2 g was intravenously administered before surgery. Blood samples were collected using a rich sampling design and cefazolin serum concentrations were measured using the HPLC/UV method. The pharmacokinetic population model was calculated using a nonlinear mixed-effects modeling approach, and the Monte Carlo simulation was used to evaluate the PK/PD target attainment. The population cefazolin central volume of distribution (Vd) of 4.91 L increased by 0.51 L with each 1 m^2^ of BSA, peripheral Vd of 22.07 L was reduced by 0.77 L or 0.79 L when using ECC or MiECC support, respectively, while clearance started at 0.045 L/h and increased by 0.49 L/h with each 1 mL/min/1.73 m^2^ of eGFR. ECC/MiECC was shown to be covariate of cefazolin Vd, but without relevance to clinical practice, while eGFR was most influential for the PK/PD target attainment. The standard dose of 2 g was sufficient for PK/PD target attainment throughout surgery in patients with normal renal status or with renal impairment. In patients with augmented renal clearance, an additive cefazolin dose should be administered 215, 245, 288 and 318 min after the first dose at MIC of 4, 3, 2 and 1.5 mg/L, respectively.

## 1. Introduction

Cardiac surgery is associated with a high risk of surgical complications. Deep sternal wound infection is one of the most serious, related with a high morbidity and mortality rate and a longer hospital stay. Antimicrobial prophylaxis is a cornerstone of its prevention, of course in combination with other measures, particularly correct indications and an adequacy of surgical techniques, thereby controlling the patient’s risk factors as well as the operating theatre environment and medical equipment [1].

The choice of antibiotic prophylaxis regimen should always depend on local epidemiologic data. However, it is well known that the main pathogens causing infections in this type of surgical wound are *Staphylococcus aureus* and coagulase-negative staphylococci, including *Staphylococcus epidermidis* [2]. The prophylactic antibiotic used must be effective against these pathogens in particular. In addition, it must have appropriate pharmacokinetic properties, especially good tissue penetration and a sufficiently long elimination half-life to ensure effective concentrations in the tissues at the site of surgery for the duration of the surgical procedure. These facts are considered by the current guidelines, based on which the most commonly used antibiotics in thoracic surgery antimicrobial prophylaxis are first-generation cephalosporins, preferably cefazolin [3].

Cefazolin is a beta-lactam antibiotic with a time-dependent antibacterial effect. In the average population, it is 70–86% bound to plasma proteins and its volume of distribution (Vd) is approximately 11 L/1.73 m^2^. It is mainly excreted by glomerular filtration and partially by tubular secretion with a mean elimination half-life of 1 h 35 min; therefore, its pharmacokinetics vary with the patient’s current state of renal function [4].

Cefazolin shows a relatively good safety profile. Among the side effects, non-serious and transient gastrointestinal discomfort may occur most often (1–10%). Less frequently (0.1–1%), hypersensitive reactions or neurotoxicity (e.g., seizures, dizziness, malaise, or fatigue) may occur. Neurological disorders are mainly associated with inappropriately high doses in patients with renal impairment, and therefore caution is needed especially in the elderly population [4].

In the cardiac surgery antimicrobial prophylaxis, the administration of an antibiotic within one hour of the skin incision is recommended. The dose of cefazolin for a patient weighing more than 60 kg should be 2 g. Additional doses are recommended every 3 to 4 h during surgery when an operation is proceeding with an open wound beyond that period [3].

However, it should be emphasized that the generally reported pharmacokinetic parameters of antibiotics are obtained in Phase I clinical trials, i.e., in a small population of healthy volunteers. It is not surprising that pharmacokinetic parameters in a population of patients undergoing cardiac surgery may vary considerably. One of the main differences characterizing these patients is the use of cardiopulmonary bypass during surgery. Over the past decades, the use of extracorporeal circulation (ECC) has become the gold standard for most cardiac surgical procedures, especially the most frequent coronary artery bypass grafting, valve repair or replacement and aortic surgery [5]. On the other hand, use of ECC causes systemic inflammatory response syndrome (SIRS), mainly caused by the contact of blood with air and foreign surfaces [6]. Clinical studies have shown that the inflammatory response to cardiopulmonary bypass (CPB) can have an unfavorable effect on clinical outcomes, albeit to a limited degree. [7].

Especially for this reason, the technology of the so-called minimally invasive ECC (MiECC) has been introduced in recent years. MiECC is particularly characterized by the use of an optimally biocompatible blood pump, minimal tubing length, separation of shed blood and exclusion of activated blood components, closed system to avoid blood–air contact, temperature management, the use of modern concepts of myocardial protection, the possibility of safe de-airing and the use of the modern concepts of fast-track anesthesia [8]. MiECC is distinguished by more stable hemodynamics during and shortly after perfusion and better end-organ protection [9].

The use of CPB may also have some effect on the pharmacokinetics of an antibiotic administered as part of antimicrobial prophylaxis immediately before surgery. According to an extensive review by Paruk et al., at least three different ways by which CPB influences its pharmacokinetics may be identified [10]. CPB may increase the distribution volume of an antibiotic by circuit sequestration, hemodilution, non-pulsatile flow and the presence of systemic inflammation (SIRS). Clearance might be increased again by the hyperdynamic state during SIRS and altered protein binding. These two factors will decrease antibiotic plasma concentrations and probably also its effect. On the other hand, hypothermia, lung isolation and organ dysfunction during CPB may decrease the clearance, resulting in higher plasma concentrations and possible adverse effects.

As for cefazolin itself, although it is the most commonly used prophylactic antibiotic in cardiac surgery, information on its pharmacokinetics during ECC remains scarce. Three older papers describe only individual pharmacokinetic data [11,12,13], as well as the paper by Kosaka et al., which focuses on the influence of renal function on its pharmacokinetics [14]. A more recent study by Lanckohr et al. found a significant influence of CPB on the Vd and elimination of cefazolin with the early repetition of 2 g cefazolin shortly after the start of CPB [15]. A population pharmacokinetic study by Asada et al. revealed decreased cefazolin clearance and an increased volume of distribution during CPB with insufficient probability of target attainment (PTA) for pathogens with MIC > 8 mg/L [16], which is fortunately not a very common situation in cardiothoracic surgery practice. Cefazolin pharmacokinetics in patients receiving MiECC support was also preliminary explored [17]; however, differences in the influence of cefazolin pharmacokinetics between ECC and MiECC—if they exist—have not yet been studied using a direct comparison or a comparison with an internal control.

We therefore aimed to develop a population pharmacokinetic model of prophylactically administered cefazolin in patients undergoing cardiac surgery in a sufficiently powered design for exploring the possible effect of both existing types of cardiopulmonary bypass: standard and minimally invasive extracorporeal circulation. The secondary aim was to propose cefazolin dosing optimization based on this model.

## 2. Results

### 2.1. Study Population and Laboratory Assays

There were 65 patients enrolled in this study. Twenty-five patients (19 males, 6 females) received MiECC support, 20 patients (19 males, 1 female) received ECC support, while 20 (17 males, 3 females) were without this modality. The demographic and laboratory characteristics of patients are summarized in Table 1. There were no significant differences in the demographic and laboratory characteristics between the study groups (patients with MiECC, patients with ECC and patients without this support). The mean duration of surgery was shorter without the use of CPB (135 min; 81–240 min) than with the use of both ECC (198 min; 155–259 min) and MiECC (191 min; 129–384 min). The average duration of CPB was 124 (70–184) min in standard ECC and 71 (45–110) min in MiECC. A total of 1296 laboratory analyses of 461 collected patients’ samples were performed to obtain plasma concentrations of cefazolin at the specified times.

### 2.2. Population Pharmacokinetic Analysis

The two-compartmental model with first-order elimination kinetics from the central compartment best fit the cefazolin concentration–time data. A proportional error model was the most accurate for the description of residual and interpatient variability. The PK model was parametrized in terms of clearance (CL), the volume of the central compartment (Vd1), the volume of the peripheral compartment (Vd2), and inter-compartmental clearance (Q). The population PK estimates and bootstrap results for the final cefazolin model are summarized in Table 2. Among the investigated variables, the most appropriate covariates were eGFR for cefazolin CL, BSA for cefazolin Vd1 and MiECC/ECC for cefazolin Vd2. The relationships between the cefazolin pharmacokinetic parameters and its covariates can be mathematically described as follows:

The diagnostic GOF plots for the final cefazolin covariate model did not show major deviations (Figure 1 and Figure 2). As shown in Table 2, the R.S.E. values (all below 50%) revealed that the parameters in the cefazolin model were precisely estimated. The population parameter estimates were similar to the median parameter values in the bootstrap procedure, indicating the reliability of the final population model. The VPC plot of the final cefazolin model revealed that the prediction was consistent with the observation, confirming the validity of the population model for the concentration–time data (Figure 3).

### 2.3. Monte Carlo Simulations

Table 3 compares the PTA values in the simulated population with the distribution of functional renal status corresponding to the real study population (eGFR of 0.67–2.8 mL/s) and to the simulated populations with eGFR of < and ≥2.17 mL/s (boundary of augmented renal clearance) at various MIC values in terms of reaching the PK/PD target of fT > MIC = 100% for at least 407 min after dose administration (data available up to 407 min after dosing). If we consider dosing to be successful when PTA is ≥90%, then the standard cefazolin dose of 2000 mg IV before surgery is sufficient at MIC values up to 4 mg/L except in patients with augmented renal clearance. On the contrary, in patients with augmented renal clearance, this dosing can be applied only if the MIC is 1 mg/L or less. Figure 4 shows simulated cefazolin concentration–time profiles after the administration of the standard dose of 2000 mg IV for the duration of surgery in patients with different functional renal statuses (eGFR < 1 mL/s, 1–2.17 mL/s and >2.17 mL/s) as the main covariate of cefazolin clearance. These simulations also showed that in patients with normal renal status or with renal impairment, the standard dose is adequate for achieving the PK/PD target throughout surgery (data available up to 407 min after dosing) in the case of MIC up to 4 mg/L. On the other hand, in patients with augmented renal clearance, an additive cefazolin dose should be administered in time depending on the MIC value. To ensure that at least 90% of the population with augmented renal clearance maintains cefazolin levels above the MIC, an additional cefazolin dose would need to be administered 215, 245, 288 and 318 min after the first dose at MIC values of 4, 3, 2 and 1.5 mg/L, respectively.

## 3. Methods

### 3.1. Study Design and Ethics

This study was designed as a prospective, observational, open-label (laboratory-blinded), pharmacokinetic trial, conducted at University Hospital in Olomouc, Czech Republic. Adult patients undergoing cardiac surgery both with and without CPB (standard or mini-invasive) were enrolled. Exclusion criteria were serum creatinine >200 µmol/L or chronic dialysis, body mass index <17 or >35 kg/m^2^, hepatic injury with an elevation of serum liver enzymes above 3× upper limit of normal, previous cefazolin administration in the 3 days preceding the prophylactic administration and the use of extracorporeal elimination methods during the surgery.

The study was performed in concordance with the ethical principles of the Declaration of Helsinki. The approval of the Ethics Committee of the University Hospital Olomouc was obtained before the study was initiated (approval No. 17-31540A). Each patient provided written informed consent before inclusion in the study.

### 3.2. Cardiopulmonary Bypass

#### 3.2.1. Standard Extra-Corporeal Circulation (ECC)

Standard CPB was carried out using the model S5 Stockert non-pulsatile roller pump system (Stockert S5, Sorin group) with oxygenator Terumo FX 25 (hard-shell reservoir, integrated arterial filter). The priming solution contained 200 mL 20% mannitol, acetated Ringer’s solution and colloids as needed. CPB was applied with a hematocrit close to 25%, under hypothermia to 34–35 °C for difficult procedures and whole-body heparinization with 300 U/kg. The pump flow was managed to ensure over 65% systemic venous oxygen saturation and an average flow rate 2.4–2.6 L/min/m^2^. A 3:1 mixture of St. Thomas cardioplegic solution (St. Thomas, Adreapharma, Czech Republic) with 800 mmol/L of additional KCl and 600 mmol/L of MgSO4 and blood was infused into the coronary arteries at an initial dose of 10 mL/kg and then at a maintenance dose 5 mL/kg every 30 min. A BC 140 plus hemoconcentration circuit (MAQUET), consisting of a polyethersulfone dialysis membrane, was used to remove the excess fluid in the circuit during CPB, as needed.

#### 3.2.2. Minimally Invasive Extra-Corporeal Circulation (MiECC)

This type of CPB was performed with a model S5 Stockert non-pulsatile centrifugal (Sorin Revolution CP) pump system (Stockert S5, SCP, Sorin group) with oxygenator Terumo FX 15 (bubletrap, without cardiotomic suction). The priming solution contained 200 mL 20% mannitol and acetated Ringer’s solution. CPB was performed with a hematocrit near 28%, normothermia 36–37 °C and whole-body heparinization with 150 U/kg. The pump flow was managed to ensure over 65% systemic venous oxygen saturation and a flow rate at 2.4–2.6 L/min/m^2^. A 3:1 mixture of St. Thomas cardioplegic solution (St. Thomas, Adreapharma, Czech Republic) with 800 mmol/L of additional KCl and 600 mmol/L of MgSO_4_ and blood was infused into the coronary arteries at a loading dose of 10 mL/kg and then repeatedly at a maintenance dose of 5 mL/kg every 30 min.

### 3.3. Cefazolin Administration and Sample Collections

An intravenous bolus of 2 g of cefazolin (Azepo 1 g, Medochemie Bohemia, spol. s r.o., Praha, Czech Republic) was administered to each patient 60–30 min before the start of surgery. Blood samples were taken at 15, 30, 45, 60, 120 and 180 min after cefazolin administration and at the end of the surgery. Three milliliters of whole blood were drawn into a vacuum tube with lithium heparin and kept on ice. Samples were taken to the laboratory immediately after surgery and centrifuged to obtain plasma. The plasma was stored at −80 °C until the analysis was performed.

### 3.4. Cefazolin Assays

The volume of 110 µL of human blood plasma was treated with 20 µL of a solution containing 1 mM metronidazole (MET, an internal standard). Subsequently, 20 μL of ultrapure water were added to the treated plasma. Plasma samples were mixed with 750 µL of methanol. After separation of the precipitated proteins, 900 µL of supernatant was evaporated to dryness under a stream of nitrogen at 37 °C. The residue was reconstituted in 100 µL of 25 mM ammonium acetate, pH 6.2. An amount of 10 µL of the prepared sample was injected into HPLC.

All measurements were performed with the Prominence LC–20A HPLC system, and an SPD-20A UV–Vis detector (Shimadzu, Kyoto, Japan). Mobile phase A consisted of 25 mM ammonium acetate, pH 6.2 and mobile phase B consisted of methanol/acetonitrile (75:25, *v*/*v*); final mobile phase A:B, 79:21, *v*/*v*. Flow rate was 0.4 mL/min. Both compounds were separated at 30 °C on Luna^®^ Omega Polar C18 column (50 × 4.6 mm; 5 µm) purchased from Phenomenex (Torrance, CA, USA). The detection was performed at 272 nm (CEF) and 320 nm (MET).

The bioanalytical assay was performed in accordance with the European Medicine Agency guidelines and has been previously described in more detail [18,19,20].

### 3.5. Population PK Analysis

Cefazolin serum concentration–time profiles were analyzed using a nonlinear mixed-effects modeling approach. The Stochastic Approximation Expectation Maximization algorithm was used to estimate the model parameters by maximum likelihood in Monolix Suite software, version 2021R1 (Lixoft SAS, Antony, France), assuming the log-normal distribution of the parameters. The model was developed in three steps.

#### 3.5.1. Base Model

One- and two-compartment models with linear and Michaelis–Menten elimination kinetics were tested for the structural model. Log-normally distributed inter-individual variability terms with estimated variance were tested on each PK parameter. Additive, proportional, and combination error models were explored for the residual error model. The selection of the most appropriate model was based upon the changes in the minimum objective function value (OFV), adequacy of the goodness-of-fit (GOF) plots, and low relative standard error (R.S.E.) values of the estimated PK parameters.

#### 3.5.2. Covariate Model

Sex and ECC/MiECC modality were tested as categorical covariates, while body weight, height, BSA, age, serum level of bilirubin, albumin, total protein, urea and creatinine, and eGFR according to the CKD-EPI formula were tested as continuous covariates. A preliminary univariate association using Pearson’s correlation test of the effects of covariates on PK estimates was made and covariates with *p* < 0.05 were considered for the covariate model. Afterwards, a stepwise covariate modeling procedure was performed. For model selection, a decrease in OFV of at least 3.84 points between nested models was considered statistically significant (*p* < 0.05 based on the χ2 test). Additional criteria for model selection were reasonably low R.S.E. values of the model parameter estimates, physiological plausibility of the parameters obtained, and GOF plots without significant bias.

#### 3.5.3. Model Evaluation

Model adequacy was evaluated using GOF plots. Observation values were plotted against individual and population prediction values. The individual-weighted residuals (IWRESs), population-weighted residuals (PWRESs) and normalized prediction distribution errors (NPDEs) were plotted against the predicted concentration plots and against the time after the dose to evaluate for randomness around the line of unity. The visual predictive check (VPC) was performed to evaluate the predictive accuracy of the final model. To build the VPC, a total of 1000 original dataset replicates were generated using the final model parameter estimates, and the simulated distribution was compared with that from the observed data. The 10th, 50th and 90th prediction percentiles of the model simulations along with their 90% confidence interval (CI) were calculated from all replicates and presented graphically.

A bootstrap analysis was performed to evaluate the stability of the model parameter. The original dataset was replicated 500 times, and the parameter estimates for each of the 500 samples were re-estimated using R package Rsmlx for Monolix Suite (Lixoft SAS, Antony, France) in the final model. The median and 95% CI obtained for each parameter estimated for the bootstrap samples were compared with the estimates in the final model.

### 3.6. Monte Carlo Simulations

Monte Carlo simulations (500 replicates of all the individuals in dataset) based on the final cefazolin pharmacokinetic model were performed to generate the theoretical distribution of pharmacokinetic profiles using Simulx version 2021 (Lixoft SAS, Antony, France).

A standard dosing regimen of 2000 mg IV was simulated for patients with different functional renal statuses (eGFR < 1 mL/s, 1–2.17 mL/s and >2.17 mL/s). The time above the minimal inhibitory concentration throughout the dosing interval (fT > MIC = 100%) was considered as the PK/PD target. The probability of target attainment (PTA) was calculated for different MIC values (0.25, 0.5, 1, 1.5. 2, 3 and 4 mg/L). For simulations, the dosing regimen was regarded to be successful if ≥90% of patients reached the PK/PD target.

### 3.7. Statistical Analyses

Median and interquartile range (IQR) were calculated using MS Excel 2016 (Microsoft Corporation, Redmond, USA). The demographic and laboratory characteristics of patients with ECC, MiECC and without this support were compared using the Kruskal–Wallis test. All comparisons were performed using GraphPad Prism 8.2.1 software (GraphPad Inc., La Jolla, CA, USA) and *p*-values of <0.05 were considered statistically significant.

## 4. Discussion

It is evident from the demographic characteristics of our study cohort that patients undergoing cardiac surgery differ from the population average in several basic demographic parameters. They are characterized by an older age and a significant male sex predominance. Overweight and reduced renal function can also be observed in some patients. Therefore, pharmacokinetic data generated from other cohorts cannot simply be applied to this population.

In our population PK model, we described BSA for central Vd, ECC or MiECC for peripheral Vd and eGFR for CL as significant covariates reducing the unexplained variability in cefazolin PK. The cefazolin population central Vd started at a baseline of 4.91 L and increased by 0.51 L with each 1 m^2^ of BSA. This means, e.g., a central Vd of 5.79 L in a patient with a BSA of 1.73 m^2^. This value fully corresponds to the observations of other authors, who report the mean value of central Vd of 5.73 L [21]. The cefazolin population CL of 0.045 L/h increased by 0.49 L/h with each 1 mL/s/1.73 m^2^ of eGFR, which is also consistent with the fact that ceftazidime is excreted unchanged into the urine mainly by glomerular filtration [4]. The population peripheral Vd of 22.07 L was reduced by 0.77 L when using ECC support or by 0.79 L when using the MiECC modality. This is a surprising observation, as logically one would rather expect an increase in Vd when using these modalities, and some other studies have reported increased Vd during ECC support [16]. Of course, it cannot be ruled out that this observation is only by chance; nevertheless, given the rich sampling scheme before, during, and after ECC/MiECC and the inclusion of patients with ECC, MiECC, and without this support, we consider our study sufficiently powered to capture the potential impact of these modalities. In any case, cefazolin belongs to the time-dependent antibiotics, in which CL plays a key role in achieving the PK/PD target, and therefore covariates of Vd are not of such clinical importance. Moreover, the observed reduction in Vd by only 3.5% would also not reach clinical manifestation.

Based on the Monte Carlo simulations, the standard cefazolin dose of 2000 mg before surgery was sufficient for PK/PD target attainment at MIC values up to 4 mg/L in patients with normal function renal status or with renal impairment, while in patients with augmented renal clearance (eGFR > 2.17 mL/s/1.73 m^2^), this dosing was sufficient only if the MIC was up to 1 mg/L. To ensure that the majority (>90%) of patients with augmented renal clearance reach the PK/PD target, an additional cefazolin dose would need to be administered 215, 245, 288 and 318 min after the first dose at MIC values of 4, 3, 2 and 1.5 mg/L, respectively.

In the simulations, we aimed for 100% time above the MIC as the PK/PD target. This target is maybe overstated for surgical prophylaxis, but it reliably reduces the risk of postoperative wound infection, which is crucial to the overall outcome. We consider the MIC of 4 mg/L as a maximal level of susceptibility of bacterial strains targeted by cefazolin in surgical prophylaxis, while the MIC of only 0.25 mg/L was the most frequent value in our study.

Our study was limited by the fact that we only assessed the PK/PD target achievement and not the real clinical outcomes. Given its controversy, the finding that ECC/MiECC reduces the peripheral Vd of cefazolin needs to be confirmed/refuted by further studies, despite its irrelevance to clinical practice.

## 5. Conclusions

Although both ECC and MiECC were shown to be covariate of the volume of distribution of cefazolin, we do not expect this finding to have a significant impact on clinical practice. As a covariate of cefazolin clearance, eGFR is of the greatest importance for the attainment of the PK/PD target. The standard dose of 2 g maintains the PK/PD target throughout surgery (up to approximately 6.5 h) in patients with normal renal status or with renal impairment. On the other hand, in patients with augmented renal clearance, an additive cefazolin dose should be administered 215, 245, 288 and 318 min after the first dose at MIC values of 4, 3, 2 and 1.5 mg/L, respectively, and thus a confirmatory trial to develop new dosing guidelines in this subpopulation would be needed.

## Figures and Tables

**Figure 1 antibiotics-11-01582-f001:**
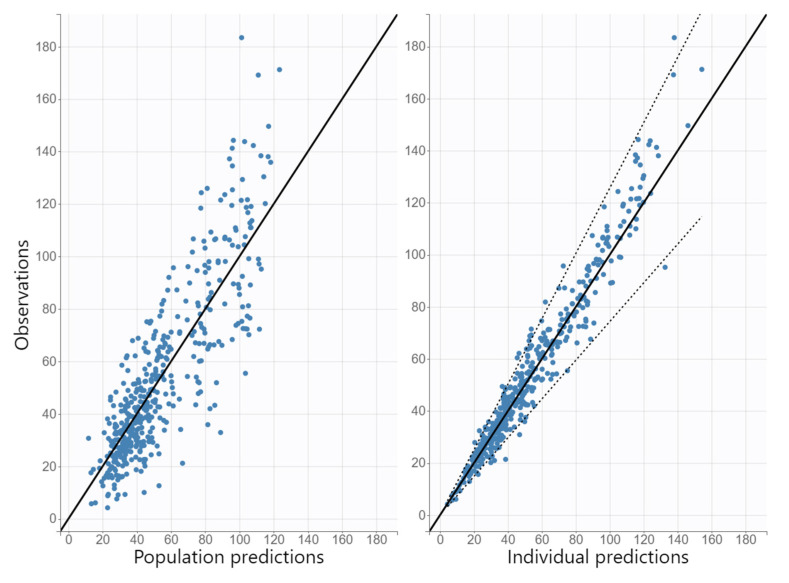
Goodness-of-fit plots obtained from the final population pharmacokinetic model for cefazolin: population and individual predictions of cefazolin versus observed concentrations.

**Figure 2 antibiotics-11-01582-f002:**
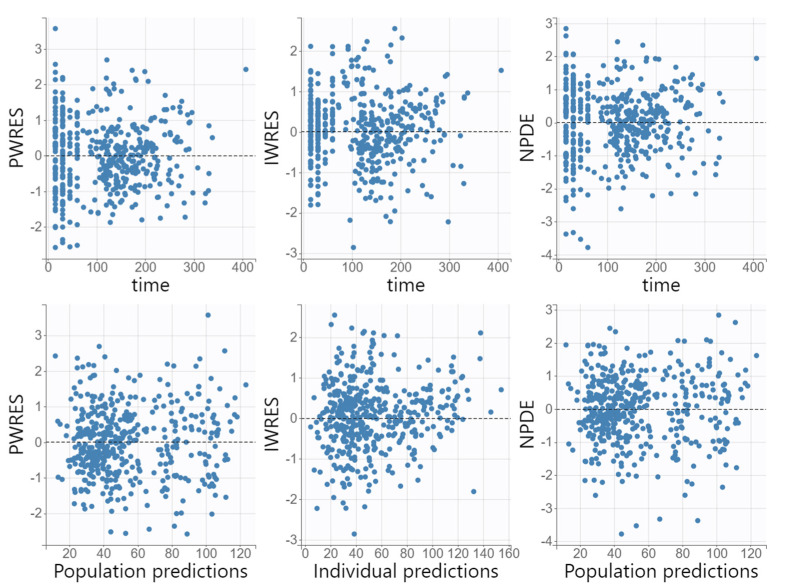
Goodness-of-fit plots obtained from the final population pharmacokinetic model for cefazolin: population-weighted residuals (PWRESs), individual-weighted residuals (IWRESs) and normalized prediction distribution errors (NPDEs) versus time after cefazolin dose and versus predicted concentrations.

**Figure 3 antibiotics-11-01582-f003:**
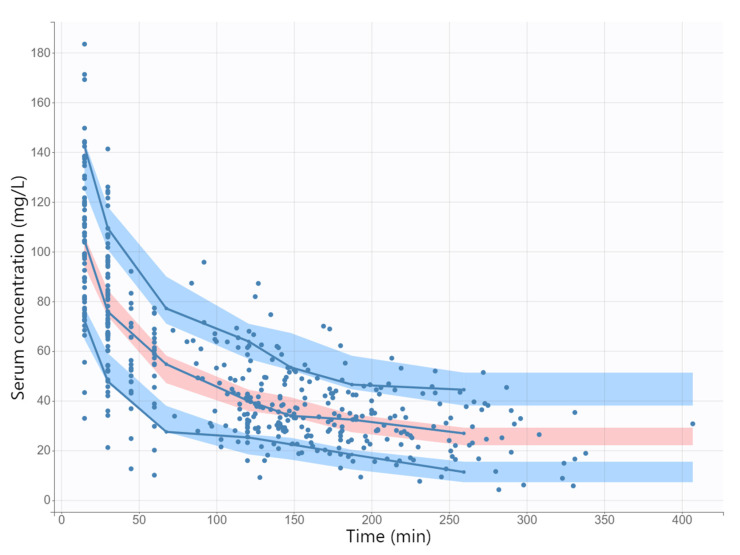
Visual predictive check (shaded areas) and observed data (circles) for cefazolin concentration versus time based on 1000 Monte Carlo simulations. Solid blue lines represent the 10th, 50th, and 90th percentiles of the observed data. Shaded regions represent 90% confidence interval around the 10th (below blue region), 50th (pink region), and 90th (above blue region) percentiles of the simulated data.

**Figure 4 antibiotics-11-01582-f004:**
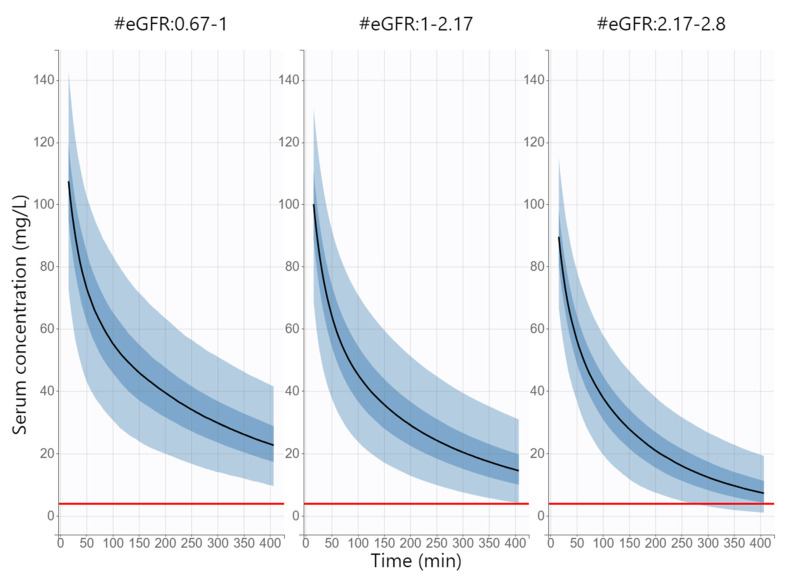
Simulated serum concentration versus time profiles after intravenous administration of 2000 mg of cefazolin to patients with different functional renal status (eGFR < 1 mL/s, 1–2.17 mL/s and >2.17 mL/s). Blue line represents the median, and the four blue bands represent percentiles (5–27.5%, 27.5–50%, 50–72.5% and 72.5–90%) of the 90% simulated concentrations distribution. Red line represents the maximal considered MIC of 4 mg/L.

**Table 1 antibiotics-11-01582-t001:** Demographic and laboratory characteristics of patients enrolled in the study.

	Non-CPB Group	MiECC Group	ECC Group	*p*-Value
Age (years)	72 (53–86)	73 (53–85)	68 (44–82)	0.4572
Body weight (kg)	86 (51–117)	89 (56–117)	85 (71–107)	0.8510
Height (cm)	171 (158–194)	172 (156–196)	176 (168–185)	0.0571
BSA (m^2^)	1.98 (1.62–2.46)	1.99 (1.58–2.37)	2.01 (1.81–2.28)	0.9202
Serum creatinine (µmol/L)	84 (50–188)	83 (57–130)	90 (64–145)	0.3512
eGFR (mL/s/1.73 m^2^)	1.36 (0.48–1.60)	1.52 (0.65–2.52)	1.36 (1.68–1.85)	0.2123
Urea (mmol/L)	4.9 (1.5–12.2)	4.8 (3.0–8.9)	5.3 (4.1–9.4)	0.5159
Total protein (g/L)	70.6 (60.3–81.0)	73.5 (59.4–84.3)	70.2 (60.7–80.1)	0.2509
Albumin (g/L)	45.3 (35.2–50.7)	45.0 (39.2–51.0)	44.0 (35.1–50.8)	0.4914
Bilirubin (µmol/L)	12.5 (5.0–50.0)	11.0 (6.0–51.0)	12.0 (4.0–43.0)	0.8111

Data are expressed as median (range). BSA is body surface area estimated according to DuBois formula. eGFR is glomerular filtration rate estimated according to CKD-EPI formula. CPB is cardiopulmonary bypass. ECC is extracorporeal circulation. MiECC is minimally invasive extracorporeal circulation.

**Table 2 antibiotics-11-01582-t002:** Estimates of the final cefazolin population pharmacokinetic model and bootstrap results based on 500 simulations.

	Final Model		Bootstrap Results	
Parameter	Estimate	R.S.E. (%)	Median	95% CI
Fixed effects				
CL_pop (L/h)	0.045	21.7	0.045	0.022–0.061
β_CL_eGFR	0.49	24.5	0.49	0.34–0.76
Vd1_pop (L)	4.91	47.6	3.87	0.76–10.20
β_Vd1_BSA	0.51	44.8	0.62	0.19–1.32
Q_pop (L/h)	0.28	17.7	0.30	0.15–0.44
Vd2_pop (L)	22.07	18.2	22.52	15.50–40.89
β_Vd2_MiECC	−0.79	29.8	−0.79	−1.33–(−0.20)
β_Vd2_ECC	−0.77	32.2	−0.74	−1.29–(−0.24)
Standard deviation of the random effects				
Ω_CL	0.35	16.8	0.33	0.21–0.48
Ω_Vd1	0.10	35.7	0.18	0.08–0.32
Ω_Q	0.77	15.6	0.66	0.33–0.94
Ω_Vd2	0.58	15.3	0.52	0.17–0.76
Error model parameters				
Proportional	0.16	4.26	0.15	0.14–0.17

Vd1 is volume of central compartment. Vd2 is volume of peripheral compartment. CL is clearance. Q is inter-compartmental clearance. BSA is body surface area estimated according to DuBois formula: Log (CL) = log (CL_pop) + β_CL_eGFR × eGFR + η_CL; Log (Vd1) = log (Vd1_pop) + β_Vd1_BSA × BSA + η_Vd1; Log (Q) = log (Q_pop) + η_Q; Log (Vd2) = log (Vd2_pop) + β_Vd2_MECC × MECC + β_Vd2_ECC × ECC + η_Vd2.

**Table 3 antibiotics-11-01582-t003:** Probability of target attainment (PTA) in simulated population with distribution of functional renal status corresponding with real study population (eGFR of 0.67–2.8 mL/s) and in simulated populations with eGFR of < and ≥2.17 mL/s (boundary of augmented renal clearance) at various minimal inhibitory concentration (MIC) values considering a PK/PD target of fT > MIC = 100% for 407 min after dose administration (based on available data).

MIC (mg/L)	eGFR = 0.67–2.8 mL/s PTA (%)	eGFR < 2.17 mL/s	eGFR ≥ 2.17 mL/s PTA (%)
0.25	99.7	99.8	97.9
0.5	99.5	99.5	96.1
1	98.8	99.0	92.4
1.5	98.1	98.3	88.7
2	97.2	97.4	84.7
3	95.1	95.5	77.1
4	92.6	93.1	69.7

eGFR is glomerular filtration rate estimated according to CKD-EPI formula. MIC is minimal inhibitory concentration.

## Data Availability

The data that support the findings of this study are available from the corresponding author upon reasonable request.

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
