# Peer review of "Population Pharmacokinetics of Prophylactic Cefazolin in Cardiac Surgery with Standard and Minimally Invasive Extracorporeal Circulation"

_antibiotics, 2022, doi:10.3390/antibiotics11111582_

Round 1
Reviewer 1 Report
The submitted manuscript entitled "Population pharmacokinetics of prophylactic cefazolin in cardiac surgery with standard and minimal invasive extracorporeal circulation" is a good work in the area of clinical study. Authors have presented a good detail regarding the use of prophylactic cefazolin. In my opinion the study is worthy and publishible in this Journal after minor suggesations.
1. Authors also have to provide the side effect of anibiotics Cefazolin on patients for all ages in introduction section.
2. Conclusion must need to elaborate with findings in their experiments, their observations and what will be the future perspectives of this work.
Author Response
The submitted manuscript entitled "Population pharmacokinetics of prophylactic cefazolin in cardiac surgery with standard and minimal invasive extracorporeal circulation" is a good work in the area of clinical study. Authors have presented a good detail regarding the use of prophylactic cefazolin. In my opinion the study is worthy and publishible in this Journal after minor suggesations.
- Authors also have to provide the side effect of anibiotics Cefazolin on patients for all ages in introduction section.
- Thank You. We added this paragraph into the introduction:
„Cefazolin shows a relatively good safety profile. Among the side effects, non-serious and transient gastrointestinal discomfort may occur most often (1-10%). Less frequently (0.1-1%), hypersensitive reactions or neurotoxicity (e.g. seizures, dizziness, malaise, or fatigue) may occur. Neurological disorders are mainly associated with inappropriate high doses in patients with renal insufficiency, and therefore caution is needed especially in elderly patients [4].“
- Conclusion must need to elaborate with findings in their experiments, their observations and what will be the future perspectives of this work.
- Thank You, we added future perspectives as follows: “…and thus confirmatory trial to develop new dosing guidelines in this subpopulation would be needed.”
Reviewer 2 Report
PopPK is extremely hot topic and I think that seeing this publication throughout the publication process is this journal’s utmost interest. However, my plagiarism detection software reported more than one third of this paper as suspicious (see the attachment).

Author Response
PopPK is extremely hot topic and I think that seeing this publication throughout the publication process is this journal’s utmost interest. However, my plagiarism detection software reported more than one third of this paper as suspicious (see the attachment).
- Thank You very much for this notice. Most of the similarities in the text are due to the repetitive use of familiar technical phrases related to the issue, especially since neither of the authors is a native English speaker. We have attempted to rephrase all passages showing higher rate of similarity. Changed passages of the text are indicated in the revised manuscript.
Reviewer 3 Report
The article shows a relevant topic for clinical application, which has been addressed by other researchers, but what is relevant is that it is applied to a geriatric population, where studies of its kind are not abundant.
Every clinical study has great limitations, and the one presented here meets the ethical requirements required by regulatory authorities, this as a preliminary observation, but which is relevant in every clinical study.
Patient management, dose management and exclusion criteria make this work valid in all aspects. The design of the experimental protocols is adjusted to those required for studies in humans, and in particular the achievement of homogeneity among the participating patients, with respect to the clinical variables, which is observed after verifying the results of biochemical analyses.
The results shown are consistent with respect to the proposed objective, and it can be mentioned that the use of the population pharmacokinetic models used in this work adhered adequately, mainly due to the proposed experimental scheme.
The authors conclude that the results observed must be verified experimentally as they have some controversies in the volume of distribution between the two experimental groups used (ECC/MiECC), but that in clinical practice this result may not be significant. The conclusion seems to me totally correct, and I agree that a perspective of the research protocol can be the review of experimental data.
Regarding the use of references, these are in accordance with the topic and with the antiquity. I don't think they have much of a problem in that regard.
Author Response
The article shows a relevant topic for clinical application, which has been addressed by other researchers, but what is relevant is that it is applied to a geriatric population, where studies of its kind are not abundant.
Every clinical study has great limitations, and the one presented here meets the ethical requirements required by regulatory authorities, this as a preliminary observation, but which is relevant in every clinical study.
Patient management, dose management and exclusion criteria make this work valid in all aspects. The design of the experimental protocols is adjusted to those required for studies in humans, and in particular the achievement of homogeneity among the participating patients, with respect to the clinical variables, which is observed after verifying the results of biochemical analyses.
The results shown are consistent with respect to the proposed objective, and it can be mentioned that the use of the population pharmacokinetic models used in this work adhered adequately, mainly due to the proposed experimental scheme.
The authors conclude that the results observed must be verified experimentally as they have some controversies in the volume of distribution between the two experimental groups used (ECC/MiECC), but that in clinical practice this result may not be significant. The conclusion seems to me totally correct, and I agree that a perspective of the research protocol can be the review of experimental data.
Regarding the use of references, these are in accordance with the topic and with the antiquity. I don't think they have much of a problem in that regard.
- Thank You very much for this positive evaluation of our manuscript. We hope that the publication of our work may inspire further research in this area and collaboration between surgeons, anaesthesiologists and clinical pharmacologists.
Round 2
Reviewer 2 Report
I find this version sufficiently improved. If the Editor agrees, this could be accepted-